# Disparities in Head and Neck Cancer: A Case for Chemoprevention with Vitamin D

**DOI:** 10.3390/nu12092638

**Published:** 2020-08-29

**Authors:** Mirela Ibrahimovic, Elizabeth Franzmann, Alison M. Mondul, Katherine M. Weh, Connor Howard, Jennifer J. Hu, W. Jarrard Goodwin, Laura A. Kresty

**Affiliations:** 1The Rogel Cancer Center, University of Michigan, Ann Arbor, MI 48109, USA; imirela@umich.edu (M.I.); amondul@umich.edu (A.M.M.); kweh@med.umich.edu (K.M.W.); conhow@med.umich.edu (C.H.); 2Department of Epidemiology, School of Public Health, University of Michigan, Ann Arbor, MI 48109, USA; 3Sylvester Comprehensive Cancer Center, University of Miami School of Medicine, Miami, FL 33136, USA; efranzman@med.miami.edu (E.F.); jhu@med.miami.edu (J.J.H.); wgoodwin@med.miami.edu (W.J.G.); 4Department of Otolaryngology, University of Miami School of Medicine, Miami, FL 33136, USA; 5Department of Surgery, Thoracic Surgery Section, University of Michigan, Ann Arbor, MI 48109, USA; 6Department of Public Health Sciences, University of Miami School of Medicine, Miami, FL 33136, USA

**Keywords:** head and neck cancer, racial disparities, vitamin D, chemoprevention, UVB, microRNA, proteomic profiling

## Abstract

Blacks experience disproportionate head and neck cancer (HNC) recurrence and mortality compared to Whites. Overall, vitamin D status is inversely associated to HNC pointing to a potential protective linkage. Although hypovitaminosis D in Blacks is well documented it has not been investigated in Black HNC patients. Thus, we conducted a prospective pilot study accessing vitamin D status in newly diagnosed HNC patients stratified by race and conducted in vitro studies to investigate mechanisms associated with potential cancer inhibitory effects of vitamin D. Outcome measures included circulating levels of vitamin D, related nutrients, and risk factor characterization as well as dietary and supplemental estimates. Vitamin D-based in vitro assays utilized proteome and microRNA (miR) profiling. Nineteen patients were enrolled, mean circulating vitamin D levels were significantly reduced in Black compared to White HNC patients, 27.3 and 20.0 ng/mL, respectively. Whites also supplemented vitamin D more frequently than Blacks who had non-significantly higher vitamin D from dietary sources. Vitamin D treatment of HNC cell lines revealed five significantly altered miRs regulating genes targeting multiple pathways in cancer based on enrichment analysis (i.e., negative regulation of cell proliferation, angiogenesis, chemokine, MAPK, and WNT signaling). Vitamin D further altered proteins involved in cancer progression, metastasis and survival supporting a potential role for vitamin D in targeted cancer prevention.

## 1. Introduction

Head and neck cancer (HNC) is the ninth most commonly diagnosed cancer in the United States and is associated with significant morbidity, mortality, and economic loss [1,2]. Important racial disparities persist for HNC [3]. Historically, HNC was more common in Black Americans than Whites, however, the incidence rate in the latter has increased since the 1990s in parallel with human papillomavirus (HPV)—positive HNC, which is diagnosed more among White males [4]. Black Americans tend to have lower rates of HPV-associated HNC which generally present as oropharyngeal cancers and are associated with better response to treatment and improved five-year survival rates compared to HPV-negative HNC [5,6]. Overall, HNC in Black patients is associated with a much lower five-year survival rate, about 30% compared to 57% in White patients [5,6]. Disparate survival rates have been investigated in recent years, yet consensus regarding the causative factors remains to be fully elucidated. Published reports document that Black HNC patients present with more advanced disease at initial diagnosis, experience greater delays in treatment initiation, and experience more gaps in health insurance coverage compared to Whites [7,8,9]. Still, our understanding of racial disparities in HNC remains incomplete. Research assessing potential biological mechanisms underlying HNC in racially diverse populations is lacking, as is information on the contribution of modifiable dietary factors to HNC incidence and progression. An inverse association between vitamin D status and cancer incidence, prognosis, and mortality has been reported in several cancer types, HNC included [10,11,12,13,14]. Moreover, it is well documented that Black Americans are at a higher risk for low vitamin D status, yet the link between HNC and vitamin D status has not been evaluated among Black patients [15,16].

Studies investigating vitamin D have identified a multitude of health benefits including improved calcium absorption, improved immune function, and an increase in bone density [17,18]. It has been speculated that vitamin D has potential anticarcinogenic effects through multiple biologic functions; however, clinical interventions and observational studies evaluating vitamin D have yielded mixed results across different cancers [10]. Nonetheless, a recent prospective cohort study targeting HNC in a population without race reported shows that higher vitamin D intake levels are associated with decreased risk of HNC recurrence [13]. Conversely, vitamin D deficiency has been linked to lymphatic metastasis among HNC patients [19].

The goal of this prospective pilot study was to assess vitamin D status in newly diagnosed Black and White HNC patients and to conduct in vitro studies to improve our understanding of mechanisms associated with the potential cancer inhibitory effects of vitamin D in the context of HNC. To our knowledge, there have been no studies evaluating the association between vitamin D levels and HNC in Black patients despite their increased risk of vitamin D deficiency and increased risk of HNC recurrence as well as reduced five-year survival rates compared to White patients. In the present analysis, we examine vitamin D levels and other characteristics in Black and White HNC patients treated for HNC in Miami, Florida. Further, to explore anti-cancer mechanisms of vitamin D microRNA (miR) and proteomic profiling was conducted on human HNC cancer cell lines following vitamin D treatment.

Our research findings support that Black HNC patients present at an earlier age and with reduced levels of circulating vitamin D levels compared to Whites even in sunny South Florida. In addition, none of our Black HNC patients had vitamin D levels greater than or equal to 30 ng/mL, the level associated with optimal regulation of parathyroid hormone (PTH), calcium absorption, and bone health. Parallel in vitro studies revealed that vitamin D treatment of HNC cell lines significantly altered miRs regulating genes involved in many cancer pathways (i.e., steroid biosynthesis, cell proliferation, angiogenesis, chemokine, stem cell pluripotency, MAPK, and WNT signaling). Similarly, proteomic profiling following vitamin D treatment revealed modulation of proteins with roles in HNC cancer progression, metastasis, chemoresistance, and cancer recurrence and further supporting a potential role vitamin D in targeted cancer prevention [20,21,22,23].

## 2. Materials and Methods

### 2.1. Characteristics of the Head and Neck Cancer Patient Population

Nineteen HNC patients (10 non-Hispanic Whites and 9 Blacks) were recruited from the Sylvester Cancer Center or Jackson Memorial Hospital in Miami, Florida. Patients 18 years of age or older were included if they had a new diagnosis of HNC cancer (including primary lesions of the oral cavity, excluding HPV-linked cancers of the oropharynx); primary lesions of stage I–IV were included. All subjects gave their informed consent for inclusion before they participated in the study. The study was conducted in accordance with the Declaration of Helsinki, and the protocol was approved by the Institutional Review Board and Ethics Committee of the University of Miami (Project ID: 81110). Both male and female patients were enrolled in the study on a rolling basis. If the participant was pregnant, suffered from other illnesses or used chronic medication known to impact vitamin D metabolism they were excluded from the study. Human subject inclusion codes were G1A, M1A, and C3A. Additional information collected included basic demographic and socioeconomic factors (i.e., age, sex, height, weight, residency, annual income, highest education level completed, risk factors, and occupational history).

### 2.2. Dietary Intake Information and Risk Factor Characterization

Block Dietary Data Systems screeners were used to collect data on the patients’ dietary intake including fruit and vegetable consumption as well as calcium and vitamin D intake. In addition to this, physical activity and sun exposure data were collected. Smoking status and alcohol consumption data were acquired by utilizing Tobacco Product Assessment Consortium (TobPRAC) tools.

### 2.3. Blood Measures

Blood levels of both the circulating metabolite (25-hydroxyvitamin D) and active metabolite (1,25-dihydroxyvitamin D) of vitamin D were measured from serum. Plasma was utilized to determine calcium and PTH levels. Levels of 25-hydroxyvitamin D and 1,25-dihydroxyvitamin D were determined using a two-step process involving rapid extraction and purification and (radioimmunoassay) RIA with specific antibodies (DiaSorin, Stillwater, MN, USA). PTH levels were determined utilizing the Immulite Immunometric assay (Diagnostic Products Corp., Los Angeles, CA, USA). Quest Diagnostics (San Juan Capistrano, CA, USA and Miramar, FL, USA) analyzed the samples utilizing well validated methods as previously reported [24]. Patients were categorized on the basis of four serum concentrations informed by the literature. 25-hydroxyvitamin D: <10.0 ng/mL, 10.0–19.9 ng/mL, 20.0–29.9 ng/mL, and >30.0 ng/mL. The first cut point is the classic definition of vitamin D deficiency and the second the level at which blood PTH homeostasis is achieved, the third a level proposed to classify an insufficient state and the fourth a level postulated for optimal regulation of PTH, calcium absorption and bone density [25]. The other two blood-based markers were assessed on a continuous scale. Students T-test were used to determine if there were statistically significant differences in blood or nutrient levels between Black and White patients. Chi-Square test was utilized to determine differences between racial groups by level and a *p*-value of 0.05 was considered statistically significant.

### 2.4. Head and Neck Cancer Cell Lines

SCC-25 (CR-1628™) and CAL-27 (CRL-2095™) cell lines were both obtained from American Type Culture Collection (ATCC). SCC-25 cells were isolated from a 70 year old White male who had squamous cell carcinoma of the tongue at T2N1 [26]. Likewise, CAL-27 cells were isolated from squamous cell carcinoma of the tongue in a White male who was 56 years old [27]. The two cell lines were plated at 3.5E06 and 1E06, respectively, and allowed to adhere for 30 h in T-75 flasks with DMEM:F12 complete medium, 10% FBS. Cells were then treated with either 2 µM of vitamin D in the form of cholecalciferol (vitamin D3, Sigma, St Louis, MO, USA) or vehicle (dilute EtOH) and harvested 24 h post-treatment utilizing a cell scraper. Cells were placed in D-PBS (without calcium or magnesium) and centrifuged at low speed (500× *g*) to pellet. The pellets were snap frozen in liquid nitrogen and stored at −80 °C until they were processed for RNA isolation, proteomic profiling, or evaluation of individual vitamin D linked proteins. The concentration of Vitamin D3 utilized for the HNSCC cell line studies were based on our preliminary viability investigations as shown (Appendix A) which show the LD50 in the range or 0.5 to 2uM as well as consideration of the published literature, both in preclinical and clinical studies [28,29,30,31,32,33]. As discussed above, circulating levels of vitamin D in humans have been targeted at >30 ng/mL (75 nmol/L or 0.075uM) [33] for optimal health. However, precise dose extrapolation from cell lines to physiological relevant human concentrations of vitamin D is complicated by the fact that levels may not accurately reflect tissue or intracellular concentrations [34]. Furthermore, human intervention studies or supplementation in free-living populations frequently involve daily vitamin D intake or in some intervention studies multiple times a day, whereas the in vitro studies herein are based on a single dose of vitamin D. Considering these factors a concentration of 2 µM was chosen for all cell-based assays.

### 2.5. RNA Isolation and microRNA Assay

RNA was isolated utilizing standard phenol-chloroform extraction procedures as previously described [35]. RNA quality was determined by Nanodrop using the 8000 Spectrophotometer (Thermo Scientific, Wilmington, NC, USA) and RNA integrity and presence of the small RNA fraction was determined using the Bioanalyzer 2100 capillary electrophoresis system (Agilent, Santa Clara, CA, USA). Sixty nanograms of total RNA was reverse transcribed using the human Megaplex Primer Pools A and B and the TaqMan miR reverse transcription kit (Applied Biosystems, Foster City, CA, USA) [35]. Each sample was pre-amplified for 12 cycles using human pool A and B Taqman^®^ Megaplex™ PreAmp Primers and PreAmp Master Mix (Applied Biosystems) and the preamplification reactions diluted, combined with TaqMan ^®^ Gene-Expression Master Mix (Applied Biosystems) divided into eight aliquots and each aliquot was added to one of the eight sample ports of the TaqMan^®^ Array A or B (v2.0), respectively. The TaqMan^®^ Array Human miR Card Set v2.0 (Thermo Fisher Scientific, Waltham, MA, USA) enables detection of 667 human miRs, 3 miR endogenous reference controls, and 1 miR assay not related to human as a negative control. Appendix A includes relevant miR platform and sequence information. The real-time PCR reactions were run according to the manufacturer’s instructions. RealTime Statminer Software (Integromics, Philadelphia, PA, USA) was used to analyze the data. The global geometric mean of all expressed miR assays was used to normalize the data. Significantly altered miRs were determined based on the most stringent criteria with a *p*-value cutoff of 1E-05 was used to determine statistically significant miRs.

### 2.6. MiR Gene Targets and Enrichment Analysis

Validated gene targets from miRs significantly altered by vitamin D treatment were determined using miRTarBase (The Chinese University of Hongkong, Shenzhen, China) [36]. The miRTarBase software and database enables users to search for validated gene targets based on the miR ID. This database was chosen as it is relatively stable and miRTarBase has been shown to be broader and more comprehensive when compared to other miR target validation databases [37]. Database for Annotation, Visualization, and Integrated Discovery (DAVID, v6.8, Frederick National Laboratory for Cancer Research, Frederick, MD, USA) was utilized to analyze the validated gene targets regulated by up or down-regulated miRs. Additionally, MetaCore software (Clarviate Analytics, Philadelphia, PA, USA) was used to further explore functional analysis of common gene targets for CAL-27 and SCC-25. *p-*values are calculated for the terms in each ontology after enrichment; the terms are then tested as separate hypotheses. The resultant *q*-value illustrates the corrected-values accounting for the total terms in the ontology including the rank of each term. This provides an estimate of the Benjamini False Discovery Rate (FDR).

### 2.7. Proteomic Profiling

Protein resuspension was achieved in 2-D cell lysis buffer (30 mM Tris-HCl, pH 8.8, containing 7 M urea, 2 M thiourea, and 4% CHAPS). This mixture was then sonicated at 4 °C followed by shaking for 30 min at room temperature, centrifugation at 4 °C (14,000 rpm) for 30 min and supernatant collection. Bio-Rads protein assay was used to measure protein concentration. The protein lysate was further processed for global proteome profiling including CyDye labeling, running of SDS gels, gel imaging, resolving protein spots, spot digestion, and MALDI-TOF MS and TOF/TOF tandem MS/MS methods were conducted in collaboration with Applied Biomics (Hayward, CA, USA).

### 2.8. Lysate Collection and Western Blot Analysis of HNC Cancer Cells Following Vitamin D

CAL-27 (7E5) and SCC-25 (7E5) cells were seeded in T-25 flasks (Corning, Thermofisher Scientific, Waltham, MA, USA) and adhered overnight prior to treatment with 2 µM vitamin D (Sigma Aldrich, Saint Louis, MO, USA) or vehicle (dilute ethanol) dissolved in phenol red free complete RPMI medium (Thermo Fisher Scientific, Scientific, Waltham, MA, USA). Cell lysates were harvested at 24 and 48 h post-treatment using RPPA lysis buffer (1% Triton X-100, 50 mM HEPES, pH 7.4, 150 mM NaCl, 1.5 mM MgCl_2_, 1 mM EGTA, 100 mM NaF, 10 mM sodium pyrophosphate, 1 mM sodium orthovanadate, and 10% glycerol) with complete EDTA-free protease and PhosSTOP phosphatase inhibitors (Sigma Aldrich, Saint Louis, MO, USA). Protein was quantified using the DC protein assay (Bio-Rad, Hercules, CA, USA). Approximately 15 µg of protein was loaded in precast 4–20% Criterion TGX gels (Bio-Rad, Hercules, CA, USA), ran for 1 h, transferred to a PVDF membrane with the Trans-Blot^®^ Turbo™ system (Bio-Rad, Hercules, CA, USA) for 30 min, blocked for 1 h at room temperature, incubated overnight with primary antibodies and incubated with the secondary antibody for 1 h. Images were captured via the ChemiDoc Molecular Imager and band quantification with ImageLab analysis software (both Bio-Rad, Hercules, CA, USA). Expression values were determined by chemiluminescent immunodetection and normalized to appropriate loading controls. Immunoblotting was performed using commercially available antibodies from Abcam (Cambridge, MA USA ): DAB2 (#ab33441; 1:500), Cell Signaling Technology (Danvers, MA USA ): GAPDH (#2118; 1:25,000), LsBio (Seattle, WA USA ): LRP2 (#LS-c667890; 1:250), Santa Cruz Biotechnology (Dallas, TX USA ): Vitamin D receptor (#sc-13133; 1:100), GAPDH (#sc-32233; 1:40,000), and Thermo Fisher Scientific, Scientific, Waltham, MA, USA): Vitamin D binding protein (#PA5-29082; 1:500).

## 3. Results

### 3.1. Patient Characteristics

Patient demographics, tobacco and alcohol use, and nutrient levels are summarized in Table 1. A total of nineteen HNC patients enrolled, 9 Black and 10 White. Black patients were significantly younger (53.6 years ± 9.4) upon HNC diagnosis compared to White patients (64.3 years ± 14.4) (*p*-value = 0.036). One hundred percent of the participants reported alcohol use. Compared to White patients, Black patients were both more likely to have a history of smoking and reported higher current smoking status. Ever smoking was reported by 66.7% of Black patients versus 30% of Whites. Among patients who smoked the average years of smoking was 24.6 years among Blacks and 34.7 years among White patients, in alignment with Whites being diagnosed at a later age. Black patients had non-significantly elevated mean body mass index (BMI) compared to Whites. All patients reported low fruit and vegetable consumption, two servings per day. Whites reported significantly higher rates of sun protection via sunscreen or protective clothing when sun exposure was ≥2 h/daily (*p-*value = 0.003, Chi-square). All Blacks reporting protecting from the sun did so by clothing or hats, not sunscreen use, whereas Whites protected with both sunscreen and apparel. Both groups reported similar sun exposure in terms of intense sun exposure, weekly sun exposure, and exposure during summer and winter months. Blacks and Whites did not differ significantly based on months living in Florida. Blacks averaged 11.0 months and Whites 10.4 months in Florida, with one Black and 3 White patients reporting living outside of Florida for at least three months. Black and White HNC patients differed significantly in terms of education, with Whites reporting significantly higher rates of college graduation as well as completing graduate or professional degrees. In contrast the highest level of education for Blacks was high school graduation, 55.6% vs. 10% among White patients. Reported annual income levels were significantly lower among Black compared to White HNC patients. Physical activity levels were low for both Black and White patients with over 70% reporting rarely or never exercising.

### 3.2. Blood Levels of Vitamin D, Parathyroid, and Calcium Among Head and Neck Cancer Patients

Table 2 reports serum and plasma measurements by patient racial group. Generally, the circulating metabolite, 25-Hydroxyvitamin D (25-OH) vitamin D is considered the most valid indicator of vitamin D status as it reflects the last 15 days; compared to the active metabolite which reflects approximately the last 15 h. The mean level of 25-Hydroxyvitamin D among Blacks diagnosed with HNC (20 ng/mL) was significantly lower than the mean among Whites (27.30 ng/mL, *p*-value = 0.04). In addition, to mean 25-Hydroxyvitamin D levels being lower in Blacks, no Black HNC patients had vitamin D levels greater than or equal to 30 ng/mL, the level associated with optimal regulation of PTH, calcium absorption and bone density. In contrast, 30% of White patients had 25-Hydroxyvitamin D levels over 30 ng/mL. About 50% of Blacks presented with 25-Hydroxyvitamin D levels in the 10–19.9 ng/mL range compared to only 10% of Whites further supporting that even in sunny South Florida vitamin D levels are lower in Blacks HNC patients compared to White HNC patients. There were no differences noted in 1,25-Dihydroxyvitamin D, which has relatively short half-life and is not considered an indicator of true vitamin D status, but rather a marker modulated only in cases of severe deficiency. Similarly, there were no differences in circulating PTH or calcium levels between the two patient populations.

### 3.3. Nutrient Levels of Vitamin D and Calcium Based on Dietary Screeners

Block Vitamin D and Calcium Screeners were utilized to gain insight into how dietary and supplement sources of vitamin D may impact circulating levels with the results summarized in Table 3. Black HNC patients reported (non-significantly) lower total vitamin D intake compared to Whites. Interestingly, compared to Whites, Black patients reported marginally higher intake of vitamin D from dietary sources (*p*-value = 0.07), whereas White patients reported non-significantly higher supplemental intake of both vitamin D and also calcium; 50% of Whites reported use of supplemental vitamin D compared to 22.2% of Black HNC patients.

### 3.4. Top miRs Dysregulated in HNC Cell Lines by Vitamin D

A total of 5 miRs were highly significantly dysregulated (*p* < 5.00E-04) in either CAL-27 or SCC-25 cell lines following vitamin D treatment as shown in Figure 1. In CAL-27 cells, hsa-miR-7-1-3p and hsa-miR-632 were significantly dysregulated by vitamin D as compared to a vehicle treated cells (*p*-value = 2.36E-05, −26.459 Fold-change and *p*-value = 9.56E-05 and 47.487 Fold-change, respectively) (Figure 1A). Vitamin D treatment of the SCC-25 cells resulted in significantly altered hsa-miR-331-5p (*p*-value = 3.82E-05, −45.008 Fold Change), hsa-miR-335 (*p*-value = 3.83E-05, −63.611 Fold Change), and hsa-miR-616 (*p*-value = 6.73E-05, 66.355 Fold Change), respectively (Figure 1A).

Next, validated gene targets regulated by the vitamin D altered miRNAs were determined using miRTarBase. The number of validated gene targets associated with significantly altered miRs in each cell line are depicted in Figure 1B. In SCC-25 and CAL-27 cells a total of 3014 and 225 validated gene targets were identified, respectively (Figure 1B). Specifically, 63 validated gene targets were identified for hsa-miR-331-5p, 2898 for hsa-miR-335, and 53 for hsa-miR-616 in SCC-25 cells, respectively. In CAL-27 cells, 184 validated gene targets were stemmed from hsa-miR-7-1 and 41 from hsa-miR-632. Figure 1B shows the total number of validated gene targets regulated by vitamin D altered miRs stratified by cell line, including 45 common or overlapping genes. A comprehensive list of validated gene targets and their respective miRs is available in Appendix A, with Appendix A (*n* = 45) and Appendix A (*n* = 32), showing the identity of overlapping genes depicted in Figure 1B–D, respectively.

Figure 1C,D illustrate the total gene targets for all significantly down and up-regulated miRs, respectively, and overlapping gene targets of miRs altered in the same direction by vitamin D treatment (detailed in Appendix A). The total 157 validated genes stem from the three up-regulated miRs which contrasts to 3082 genes regulated by the two down-regulated miRs. In Figure 1C, down-regulated miR-335 contributes the majority of presumably up-regulated gene targets compared to miR-7-1-3p. The two down-regulated miRs share 32 validated gene targets (Appendix A) with diverse cancer-associated functions. Figure 1D illustrates the lack of overlapping genes among up-regulated miRs. There is only one common gene target, POLD3, that is shared between miR-632 and miR-331-5p, but not miR-616-3p. Overall, there is a considerably reduced number of validated gene targets for the up-regulated miRs compared to the down-regulated; still, a number of resultant genes have documented roles in cell proliferation, DNA damage response, cancer stemness, extracellular matrix organization, and genome stability (Appendix A). In contrast with the down-regulated miRs, all three of the up-regulated miRs have similar numbers of validated gene targets.

### 3.5. DAVID Enrichment Analysis

Two approaches of enrichment analysis were applied. The first based on the direction of miR change and resultant genes (157 and 3082) regulated by those miRs altered by vitamin D and the second based on all miRs and resultant common genes altered in both cell lines by vitamin D treatment (*n* = 45). The genes regulated by vitamin D driven miR dysregulation were analyzed for their respective Kyoto Encyclopedia of Genes and Genomes (KEGG) pathways, biological processes, molecular function, and disease linkages. Table 4 summarizes the top most significantly altered functional terms based on all up or down-regulated miRs and subsequent validated gene targets (*n* = 157 and 3082). The top KEGG pathway for the up-regulated miRs, presumably down-regulated gene targets, is Pathways in cancer (*p*-value = 0.01, FDR = 0.90) with nine genes contributing *(CKD4, DVL3, ETS1, MSH6, XIAP, RHOA, SMAD4, CCDC6,* and *LPAR2*). Expanded results, beyond the top changes, are detailed in Appendix A. Stemming from vitamin D down-regulated miRs, the most significant identified KEGG pathway (Table 4) is Steroid biosynthesis (*p*-value = 7.72E-06, FDR = 2.2E-03, 13 genes), followed by MAPK signaling pathway (*p*-value = 3.89E-05, FDR = 5.52E-03, 65 genes), and Signaling pathways regulating pluripotency of stem cells (*p*-value = 6.80E-05, FDR = 6.43E-03, 41 genes). Additional significantly altered pathways linked to down-regulated miRs include Estrogen, PI3K-AKT, RAS, and Chemokine signaling (Appendix A).

Nine biological processes were significantly altered by vitamin D based on genes linked to down-regulated miRs including Negative regulation of cell proliferation (*p*-value = 5.27E-06, FDR = 0.03, 95 genes), Angiogenesis (*p-*value = 1.03E-05, FDR = 0.03, 60 genes), and Canonical WNT signaling (*p-*value = 2.26E-05, FDR = 0.03, 29 genes) as well as Extracellular matrix and Cell adhesion. Genes stemming from up-regulated miRs resulted in Positive regulation of transcription from RNA polymerase II promoter as the top process, but it was not significant once FDR corrected.

In terms of molecular function, vitamin D linked up-regulated miRs and subsequent genes identified only Protein binding (*p-*value = 1.50E-04, FDR = 0.04, 96 genes) as a significant function.

Down-regulated miRs and related genes revealed only Transcription factor activity, sequence-specific DNA binding (*p*-value = 1.47E-05, FDR = 0.03, 194 genes). Finally, top diseases identified based on miRs up and down-regulated by vitamin D include Cancer and Chemodependency, respectively.

In addition, enrichment analysis was conducted based on the 45 validated gene targets shared in both vitamin D treated HNC cell lines (as illustrated in Figure 1A and detailed in Appendix A) revealing Focal adhesion as the only significant pathway (*p*-value = 3.20E-04, FDR = 2.90E-02, 6 genes) following vitamin D treatment. Genes included *COL4A1, EGFR, IGF1R, ITGA1, PAK3,* and *VEGFA*, many with relevance to HNC. Similar to the results stemming from up or down-regulated miR driven genes (Figure 1C,D), Protein binding was the top molecular function identified based on the 45 shared genes in CAL-27 and SCC-25 cells.

### 3.6. Protein Level Changes in HNC Cells Following Vitamin D Treatment

Western blot results implicate response differences to vitamin D treatment between CAL-27 cells and SCC-25 cells (Figure 2). Western results show that levels of LRP2 and DAB2 proteins are constitutively lower in CAL-27 cells as compared to SCC-25 cells. The vitamin D receptor is expressed more strongly in SCC-25 as well. However, vitamin D binding protein is expressed more strongly in CAL-27 cells. Vitamin D treatment increased levels of the vitamin D receptor in both HNC cell lines, with strongest effects noted in SCC-25 cells. Vitamin D treatment modestly increased DAB2 levels only in CAL-27 cells. Vitamin D driven LRP2 increases were only observed in SCC-25 cells supporting differential responses between the cell lines. Findings may reflect that HNC cells increase their ability to endocytose vitamin D and bind vitamin D at the nuclear receptor in response to vitamin D treatment. The observed differences between cell lines may be due to molecular differences between them, raising the possibility that there may also be different molecular profiles in patient populations; expression of these proteins should be examined in patient samples. These findings may have relevance to the relative importance of free versus bound vitamin D in HNC and to the metabolism and processing of vitamin D in vivo [38,39].

Proteomic profiling was also employed as an untargeted approach to determine additional proteins modified by vitamin D utilizing of SCC-25 and CAL-27 human HNC cell lines with results summarized in Table 5. A total of six proteins were identified as highly dysregulated by vitamin D. Nucleophosmin, Lactoylglutathione lyase, Heat shock protein beta-1, and Ras-related protein Rap-2b were markedly downregulated in SCC-25 cells. In CAL-27 cells, Peroxiredoxin-1 and Histone H2A type 1-J were down and up-regulated, respectively (Table 5).

## 4. Discussion

Many studies have investigated the underlying factors that contribute to racial disparities in HNC patients pointing to a complex and multifactorial etiology. Black HNC patients have a 20–30% reduction in five-year relative survival compared to Whites [40,41]. In turn, Black Americans have higher age-adjusted HNC mortality rates compared to Whites [42]. The literature supports that Black patients are more likely to be diagnosed with higher stage disease and distant metastases, and experience increased mortality following a diagnosis of HNC [9,43,44]. Black HNC patients are also significantly less likely to undergo surgical treatment, even among patients with similar health insurance status [9,45,46]. Sociodemographic factors are frequently cited as dominate causes to cancer disparities, including HNC [47,48,49,50]. However, a recent meta-analysis including ten studies with greater than 100,000 patients reported poorer survival among Black HNC patients after controlling for socioeconomic factors as well as tumor stage and treatment variables supporting that additional factors contribute to the observed racial differences [51]. Thus, the objective of this prospective pilot study was to assess vitamin D status in newly diagnosed HNC patients stratified by race and to gain potential mechanistic insight by performing in vitro studies utilizing human HNC cell lines. Among Whites vitamin D status is inversely associated with HNC, low vitamin D levels in White HNC patients is also linked to cancer progression, increased recurrence, and metastatic disease [13,14,19]. Despite well documented hypovitaminosis D in Blacks, vitamin D status has not been assessed in HNC patients stratified by race.

Our pilot study results showed significantly lower levels of circulating vitamin D (25-hydroxyvitamin D) in newly diagnosed Black HNC patients (20.0 ng/mL) compared to the White patients (27.3 ng/mL). Although this is the first such report in Blacks with HNC, it does align with earlier research in Blacks without cancer which also reported reduced vitamin D levels in the range of 18.0–25.1 ng/mL [15,16,52,53,54,55]. Higher levels of melanin pigmentation in darker skin is known to be a contributory cause of lower levels of vitamin D among Blacks. In addition, an inverse association between BMI and vitamin D level has been noted previously, including during interventional studies [56,57,58]. Although the mean BMI was higher among Black HNC patients and more Blacks were overweight based on BMI, the differences were not statistically significant. Sun exposure is also known to impact vitamin D levels and to interact with obesity [59]. All HNC patients reported similar levels of sun exposure, but Whites reported more frequent use of sunscreen compared to Blacks. Overall, both groups of HNC patients had low levels of vitamin D, with none of the Black patients and only 30% of the White patients having levels ≥30 ng/mL, which is required for optimal PTH regulation, calcium absorption, and bone density [25]. Moreover, 50% of Black HNC patients had deficient vitamin D levels, below 20 ng/mL. Interestingly, dietary and supplement intake measures also indicated that Black HNC patients had lower total vitamin D, but they received more from dietary sources, whereas White patients supplemented more frequently. Two previous studies have also reported lower supplemental vitamin D intake in Blacks compared to Whites [60,61]. Although randomized clinical trials and interventional studies with vitamin D have had mixed results, targeting deficient populations may prove beneficial, especially in the context of racial disparities [10,62,63].

Similar to previous larger studies and published metanalysis focused on racial disparities in HNC [8,9,64,65], our pilot study results support that Black and White HNC patients differ significantly in terms of highest educational obtainment and annual income level. Exposure to other known HNC risk factors appeared similar between Whites and Blacks, including alcohol use and current smoking status. Blacks HNC patients did have non-significantly increased rates of ever smoking compared to Whites, and they were diagnosed about ten years earlier than Whites with HNC for reasons that remain unclear. In addition, Blacks who smoked, did so on average for 10 fewer years compared to Whites when diagnosed with HNC, raising the possibility that other, less well characterized factors may indeed be contributing to HNC progression and disparate survival among Blacks.

Tobacco, alcohol, and HPV are considered among the major modifiable risk factors for HNC [5]. Tobacco use has declined in the United States reducing tobacco linked HNC and in turn reducing age adjusted HNC incidence rates among Blacks [8,66]; yet, Blacks still have elevated age adjusted mortality rates for HNC compared to Whites [3,8,42,67]. In addition, HPV-positive HNC has markedly increased in recent years, but is generally more common among White males and has more favorable treatment outcomes [68,69,70,71]; thus, neither tobacco use or HPV prevalence patterns appear to fully explain the racial disparity in HNC outcomes. Although this is a small pilot study, we report differences in vitamin D levels among newly diagnosed Black and White HNC patients which may contribute to disparities in HNC outcomes. The topic merits further investigation in a larger cohort study incorporating additional relevant clinical variables such as cancer stage, grade, subsite analysis, treatment choice and survival outcomes.

In vitro studies were undertaken to explore potential mechanisms of vitamin D activity in two human HNC cell lines, CAL-27 and SCC-25, both of squamous cell origin. Vitamin D treatment significantly altered five miRs, each regulating between 41 to 2898 genes. Of the miRs modulated by vitamin D, Hsa-miR-7-1 and hsa-miR-335 were significantly down-regulated; whereas, hsa-miR-331-5p, hsa-miR-616, and hsa-miR-632 were significantly up-regulated. Enrichment analysis via DAVID was conducted based on the genes regulated by each miR within a cell line, and also separately based on genes regulated by vitamin D induced up or down-regulated miRs revealing effects on multiple cancer relevant pathways and processes. There is evidence that both down-regulated miRs have oncogenic activity in HNC or precursors lesions [72,73,74,75,76,77,78,79,80,81,82,83,84,85]. Analysis of genes regulated by down-regulated miRs revealed modulation of multiple pathways relevant to HNC including MAPK, PI3K, RAS, and Chemokine signaling. Additionally, the top biological processes altered based on genes regulated by down-regulated miRs was Negative regulation of cellular proliferation followed by Angiogenesis; Canonical WNT signaling and Cell adhesion were also among the significant processes. Two previous studies investigating the role of miR-7 in lung cancer, which shares risk factors with HNC, reported that miR-7 is induced by EGFR/Ras/ERK/Myc signaling leading to aberrant cell proliferation and migration [75,86]. Together these results support a cancer inhibitory role for vitamin D through impacting key drivers identified in HNC [87]. Considering that HPV-positive HNC are documented to have double the PI3-kinase activating mutations (50% as compared to HPV-negative ones, which have about 25%) future studies should expand evaluations to include HPV-positive models or tissues [88].

Vitamin D treatment of HNC cell lines up-regulated hsa- miR-331-5p, hsa-miR-616-3p, and hsa-miR-632. Combined, these miRs presumably down-regulate 157 gene targets, many with well documented roles in cell proliferation, signal transduction, DNA damage response, cancer stemness, adhesion, extracellular matrix (EMT) organization, and genome stability (i.e., *MDM4, CDK4, MSH6, XIAP, ETS1, RHOA, SMAD4, POLD3, COL3A1, FGF2, FOXF2,* and *SOX5*). Notably, *MDM4* plays a role in regulating p53 and has been linked to cancer recurrence and poor outcomes in HNC patients [89,90,91]. As another example, the transcription of *CDK4* is linked to tobacco mediated oral carcinogenesis and acts as a potent cyclin dependent Kinase 4 Regulatory Factor (KRF) and a potential cancer target [92]. In terms of specific miRs, vitamin D treatment increased miR-331-5p. Reports support that miR-331-5p, as well as the 3p strand, differ in patients with laryngeal squamous cell carcinoma [80,82,93]. Moreover, overexpression of miR-331-3p inhibits cell proliferation and invasion while promoting apoptosis via reduced expression of elF4B and subsequent inhibition of the phosphorylation of PI3K/AKT signaling molecules [94]. Similarly, in lung cancer, increased expression of miR-331 has cancer inhibitory effects through down-regulation of MAPK, suppression of EMT as well as inhibition of metastatic ability of cancer cells in vitro and in vivo [95,96]. Results for miR-632 in the context of HNC appear mixed. One study reported that under-expression of miR-632 in saliva from oral squamous cell carcinoma patients compared to healthy controls suggesting increasing levels by vitamin D may prove favorable [97]. Similarly, Lu et al. suggested miR-632 to be a tumor suppressor in laryngeal cancers where it is down-regulated potentially via CCR6 and p38 dependent mechanisms [98]. Conversely, another study reported that increased expression of miR-632 in laryngeal tissues and cell lines accelerates cell proliferation, migration, and invasion supporting a more oncogenic function [99]. Contradictory finding may be due to heterogeneity of cell lines and even patient samples. It is well documented that miRs can act in a cancer specific manner, but there is less research documenting site specific effects within a target or evaluating additional sources of heterogeneity. To our knowledge miR-616-3p has not been reported as dysregulated in HNC. Our data supports that vitamin D modulates a select panel of miRs in HNC cell lines, which ultimately interferes with many cancer hallmarks. Additional research has evaluated vitamin D modulation of miRs in lung, cervical, and breast cancers [66]. However, the miRs identified in previously published studies do not overlap with those identified as significantly modulated by vitamin D in HNC cell lines.

Proteomic profiling was conducted to assess whether vitamin D held potential to impact or correct known defects in protein machinery associated with HNC. In total, vitamin D markedly increased one protein and down-regulated five proteins, all with documented roles in cancer [22,100,101,102,103,104]. Nucleophosmin (NPM), is associated with evasion of apoptosis, increased cancer cell viability, growth, and cell proliferation [105], and was down-regulated following vitamin D treatment of SCC-25 cells. In a recent study increased expression of NPM was reported in 82% of laryngeal cancer tissues and NPM knockdown inhibited laryngeal cancer cell survival [23]. The full results revealed an oncogenic role for NPM in laryngeal cancer through its effects on apoptosis and cellular growth. Overexpression of NPM has been documented in oral squamous cell carcinoma (OSCC); immunohistochemistry and immunofluorescent staining showed significantly elevated expression levels in OSCC patient samples compared to control [106]. Another OSCC focused study reported NPM silencing induced genes involved in apoptosis and downregulated of procarcinogenesis genes [107]. The down-regulation of NPM following vitamin D treatment is consistent with our enrichment analysis results given NPM has a role in the Myc-ARF-p53 pathway and the molecule functions as a histone chaperone [108,109]. The latter may also explain the reduction noted in Histone H2A type 1-j (H2A1J) following vitamin D. Similarly, lactoylglutathione lyase (GLO1), was down-regulated in HNC cell lines after vitamin D treatment and is documented to be significantly overexpressed in OSCC tissues [110]. Importantly, there is recent evidence that GLO1 plays a critical role in invasion and metastasis of oropharyngeal tumors, in addition to initiation and maintenance of tumor growth. It has been reported that patients with high GLO1 expression have significantly shorter disease-specific survival [110]. GLO1 has only been characterized in a small number of HNC studies as cited; however, it is shown to have multiple roles in promoting cancer cell survival, proliferation, and is a likely target for chemotherapy based on the broader literature [111].

Vitamin D down-regulated Heat shock protein beta-1 (HSP27), a multi-functional protein with well documented roles in HNC inflammation, proliferation, cancer progression, stemness, EMT, and more recently radio-sensitization and therapeutic resistance [21,112,113,114,115,116,117]. Moreover, in SCC of the tongue HSP27 inhibition represses apoptosis and enhances sensitivity to chemotherapies [21] supporting a role for agents that reduce or impair HSP27. HSP27 is also a downstream target of the PI3K/AKT signaling pathway [112] which links our proteomic results to our enrichment analysis based on vitamin D induced miR alterations. Finally, expression levels of a number of RAB family members segregate metastatic versus non-metastatic oral cancers, including *RAB2B* [118]. In addition, the latter study reported knock-down of *RAB5, RAB7*, and *RAB11* in SCC-25 cells inhibits cancer cell migration and invasion supporting that agents downregulating RAB family members may impart cancer inhibitory potential.

Two additional proteins were altered by vitamin D treatment specifically in CAL-27 HNC cells. H2A1-J was up-regulated by vitamin D but has not been reported on in HNC. However, H2A1 depletion is linked to induction of cancer cell stemness in hepatocellular carcinoma [119] which aligns with gene enrichment analysis conducted based on miRs dysregulated by vitamin D in our study. Additionally, modulated by vitamin D in CAL-27 cells was Peroxiredoxin-1 (PRX-1). It was down-regulated, further supporting a role for vitamin D in suppressing HNC associated pathways. An evaluation of Prx-1 in human OSCC tissues showed elevated expression in OSCC samples compared to controls; interestingly investigators also saw a dose-dependent elevation in Prx-1 such that expression was highest in smokers with OSCC and lowest in control tissues [120]. Additional research has shown silencing of Prx-1 in CAL-27 and SCC-15 blocks promotion of proliferation and migration and Prx-1 has the ability to promote EMT processes via NFκB linked activity, as has been reported for NPM and HSP27 [120,121]. Thus, vitamin D treatment of HNC cell lines results in potent down-regulation of proteins implicated in multiple aspects of cancer, from inflammation, to aberrant proliferation, to migration, altered EMT, adhesion, and therapeutic resistance. Interestingly, a number of the vitamin D inhibited proteins are known to converge on a common transcription factor, NFkB which has also been proposed as a potential target for increased therapeutic efficacy in HNC [122].

## 5. Conclusions

Limitations of our research include a relatively small sample size, but this was intended to be a first assessment for determining whether a larger study was warranted. In addition, our small sample size of 19 HNC patients precludes making any linkages to sex, stage, grade, site-specific effects, or survival. Finally, to our knowledge there are no HNC cell lines available for research which are derived from Black patients; thus, our in vitro work was limited to HNC cell lines derived from White HNC patients.

Still, our data shows for the first time that circulating vitamin D levels are significantly depressed in newly diagnosed Black HNC patients compared to Whites. Furthermore, in vitro studies targeting mechanisms by which vitamin D may inhibit HNC revealed activity targeting early to late cancer related events; spanning from inflammation and chemokine changes to alterations in EMT, drivers of recurrence, and therapeutic resistance. Despite mixed results from vitamin D trials in other targets, our results support that vitamin D modulates a number of cancer pathways, biological processes, genes, and proteins with well documented roles in HNC development, response to therapy and disease recurrence which remains a significant issue. Study results support conducting future research to evaluate vitamin D in larger cohorts stratified by race and with sufficient power to interpret key clinical correlates and survival outcomes. Our results also revealed new vitamin D induced miR alterations which paralleled changes in many HNC relevant proteins paving the way for future genetic studies to interrogate miR dysregulation relative protein function. Ultimately, our results may inform and guide future in vitro, in vivo, and clinical chemoprevention studies assessing the efficacy of vitamin D as an intervention strategy for vulnerable or high risk populations, whether it be based on race or other variables imparting increased risk for HNC.

## Figures and Tables

**Figure 1 nutrients-12-02638-f001:**
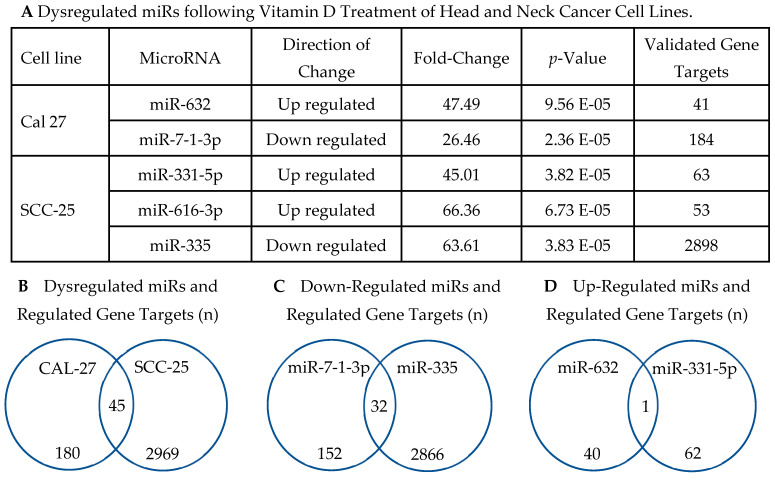
Summary of Significantly Dysregulated miRs following Vitamin D Treatment of Head and Neck Cancer Cell Lines. (**A**) Vitamin D altered miRs in CAL-27 and SCC-25 cells, the direction of change, and the number of validated gene targets regulated by each miR with 45 overlapping genes. (**B**) Validated gene targets regulated by Vitamin D modulated miRs in CAL-27 and SCC-25 cells and overlapping gene targets. (**C**) Specific miRs down regulated by Vitamin D in CAL-27 and SCC-25 cells, with overlapping genes identified (*n* = 32). (**D**) Up-regulated miRs showing individual results and overlapping gene targets (*n* = 1).

**Figure 2 nutrients-12-02638-f002:**
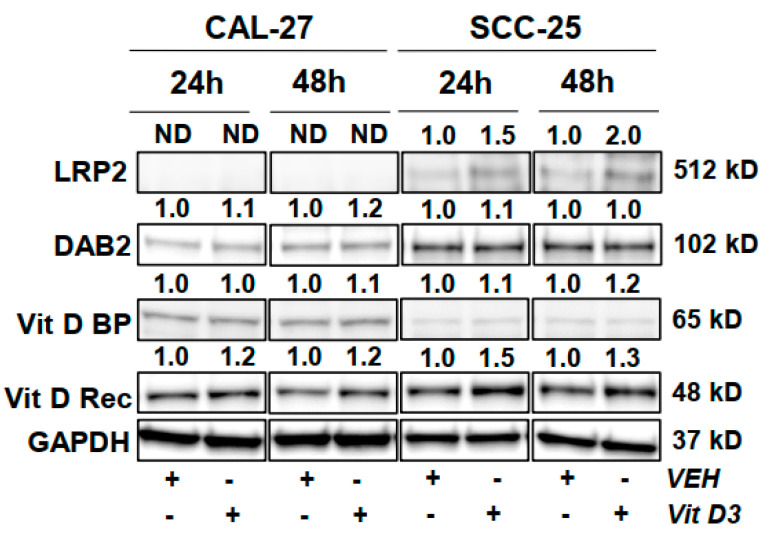
Modulation of vitamin D related proteins. Western blot results for CAL-27 and SCC-25 HNC cell lines following 2 µM vitamin D treatment. ND indicates no detection. Quantitation was normalized to GAPDH as a loading control and fold-change from vehicle reported.

**Table 1 nutrients-12-02638-t001:** Characteristics of newly diagnosed head and neck cancer patients.

Characteristics	Race	
	Black	White
Average Age (years, ±SD)	53.6 (±9.4)	64.3 (±14.4) #
Ratio of Men:Women (*n*)	5:4	8:2
Alcohol Use	100%	100%
Current Smoker	33.3%	20%
Ever Smoker	66.7%	30%
Body Mass Index (mean kg/m^2^)	27.1	24.2
Fruit & Vegetable Servings/Daily	2.0	2.1
Sun Protection, Never or Seldom	89.9%	20.0%
Sun Protection, Always or Mostly	11.1%	80.0% #
Middle School or Some High School (*n*)	22.2% (2)	0%
High School Graduate (*n*)	55.6% (5)	10% (1)
Some College (*n*)	11.1% (1)	30.0% (3)
College Graduate or Professional Degree (*n*)	11.1% (1)	60.0% (6) #

Education is reported as highest level achieved; Sun protection use refers to using clothing or sunscreen for periods of sun exposer ≥2 h a day; and # Statistically significantly different based on Students *T*-Test or the Chi-Square test statistic for population differences across categories (*p*-value < 0.05).

**Table 2 nutrients-12-02638-t002:** Blood levels of vitamin D, parathyroid, and calcium among head and neck cancer patients.

Measurement	Race		*p*-Value
	Black	White	
25-Hydroxyvitamin D (ng/mL)	20.00 (±5.98)	27.30 (±9.86)	0.04
1,25-Dihydroxyvitamin D (pg/mL)	43.86 (±20.80)	36.80 (±14.33)	0.21
Patients with Low Vitamin D (<19.9 ng/mL)	50%	10%	
Patients with Intermediate Vitamin D (20.0–29.9 ng/mL)	50%	60%	
Patients with Sufficient Vitamin D (≥30 ng/mL)	0%	30%	0.06
Parathyroid (pg/mL)	35.50 (±23.95)	35.89 (±14.93)	0.48
Calcium (pg/mL)	9.34 (0.58)	9.09 (±0.68)	0.21

*p-*value based on Students *T*-test for nutrient levels and Chi-square test statistic for vitamin D differences by concentrations in the populations (<0.05 considered statistically significant).

**Table 3 nutrients-12-02638-t003:** Nutrient levels of vitamin D and calcium based on dietary screeners.

Measurement	Race		*p*-Value
	Black	White	
Total Vitamin D (IU)	174.70 (±154.37)	276.93 (±206.03)	0.23
Dietary Vitamin D (IU)	121.24 (±113.77)	62.92 (±40.10)	0.07
Supplemental Vitamin D (IU)	88.89 (±176.38)	200.00 (±210.82)	0.12
Patients Supplementing Vitamin D	22.2%	50.0%	0.21
Total Calcium (mg)	678.40 (±450.42)	542.93 (±280.74)	0.22
Dietary Calcium (mg)	535.16 (±317.30)	447.87 (±286.89)	0.27
Supplemental Calcium (mean, mg)	25.28 (±56.10)	83.69 (±82.94)	0.31
Patients Supplementing Calcium	44.4%	60.0%	0.34

*p-*Value based on Students T-test or Chi-square test for population measures (<0.05 considered statistically significant).

**Table 4 nutrients-12-02638-t004:** Top pathways, biological processes, and molecular functions and diseases for altered MiR. Target Genes following vitamin D Treatment of CAL-27 and SCC-25 head and neck cell lines.

Functional Parameter	MiR Alteration	Term (*n*)	*p*-Value	Benjamini-Hochberg FDR
KEGG Pathway	Up-regulated	Pathways in Cancer (9)	0.01	0.90
	Down-regulated	Steroid Biosynthesis (13)	7.72E-06	2.2E-03
Biological Process	Up-regulated	Positive regulation of transcription from RNA polymerase II promoter (18)	3.31E-03	0.97
	Down-regulated	Negative regulation of cell proliferation (95)	5.27E-06	0.03
Molecular Function	Up-regulated	Protein binding (96)	1.50E-04	0.04
	Down-regulated	Transcription factor activity, sequence-specific DNA binding (194)	1.47E-05	0.03
Diseases	Up-regulated	Cancer (95)	4.30 E-04	7.71E-03
	Down-regulated	Chemodependency (736)	2.91E-12	5.53E-11

FDR: False Discovery Rate.

**Table 5 nutrients-12-02638-t005:** Proteins modified by vitamin D treatment of head and neck cancer cell lines.

Cells	Spot	Protein	Symbol	Fold-Change	Uniprot No.
SCC-25	15/16 *	Nucleophosmin	NPM_HUMAN	−2.36	P06748
	28	Lactoylglutathione lyase	LGUL_HUMAN	−2.73	Q04760
	31	Heat shock protein beta-1	HSPB1_HUMAN	−2.57	P04792
	51	Ras-related protein Rap-2b	RAP2B_HUMAN	−3.39	P61225
CAL-27	30	Peroxiredoxin-1	PRDX1_HUMAN	−2.60	Q06830
	31	Histone H2A type 1-J	H2A1J_HUMAN	+2.48	Q99878

* The fold-change for Nucleophosmin is an average of the two detected spots 15 and 16 following treatment with vitamin D.

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
