# Peer review of "Disparities in Head and Neck Cancer: A Case for Chemoprevention with Vitamin D"

_nutrients, 2020, doi:10.3390/nu12092638_

Round 1

Reviewer 1 Report

The authors have made a large manuscript, which includes patient samples, cell culture in various lines and a great job searching the results databases, which includes a large number of proteins determined by western blot and a fantastic review of the results. articles published on the subject, and it is a very complete work and with a beautiful result, which has few points to improve. They have been able to summarize the results very well, although in the revision document that I received, the supplementary materials did not appear, and it would be interesting to see them in the new version that requires a minor revision with the following points.
1.- Include the supplementary material, only in the case of not having received it, since in the version to review only the web address of the manuscript appears, which does not work at the moment.
2.- The approval code of the procedures used in point 2.1 of material and methods could be included.

3.- In point 2.5, the sequences of the primers used according to the MIQE standard should be referenced as supplementary material.

4.- As a curiosity, in the section on level of studies in Table 1, they do not coincide with the n for each group. What is the reason?

5.- In the same table, the data on the use of sun protection appears in% in white but not in black.

Just those minor modifications. Congratulations to the authors for the manuscript.

Reviewer 2 Report

Thank you for the opportunity to review this manuscript for Nutrients. The authors present a two-part study examining vitamin D in head and neck cancer. The first part, an epidemiologic/health disparities investigation into vitamin D levels in head and neck cancer patients of different racial backgrounds seems to clearly indicate that the darker skinned population have lower levels of vitamin D.  The second part investigated the effects of vitamin D treatment versus vehicle on two head and neck cancer cell lines and studied numerous cancer pathways. The authors report treating the treatment group with to micromolar vitamin D. Above in the epidemiologic study vitamin D levels are reported in nanograms per mL, the more clinically conventional measure. It seems that the cells were exposed to approximately 350 ng per ml vitamin D3, by my calculation. This is above what would clinically be considered a toxic level, however it is clinically undetermined whether or not this type of a dose could be considered in the head and neck cancer patients. It might be helpful if the authors were to more clearly describe why they chose this dose and what the potential chemotherapeutic effects and side effects could be, were an equivalent tissue concentration achieved in patients with head and neck cancer. This type of an explanation may connect the two pieces of this otherwise very well written and polished manuscript which currently seem to be less well related to each other than they could be.  
